# Climate Change Stressors, Phosphate Limitation, and High Irradiation Interact to Increase *Alexandrium minutum* Toxicity and Modulate Encystment Rates

**DOI:** 10.3390/microorganisms12071480

**Published:** 2024-07-19

**Authors:** Marta Sixto, Pilar Riobó, Francisco Rodríguez, Patricio A. Díaz, Rosa I. Figueroa

**Affiliations:** 1Centro Oceanográfico de Vigo, Instituto Español de Oceanografía (IEO-CSIC), Subida a Radio Faro 50-52, 36390 Vigo, Spain; francisco.rodriguez@ieo.csic.es; 2Campus do Mar, Facultad de Ciencias del Mar, Universidad de Vigo, 36311 Vigo, Spain; 3Instituto de Investigaciones Marinas, Consejo Superior de Investigaciones Científicas (IIM-CSIC), Eduardo Cabello 6, 36208 Vigo, Spain; pilarriobo@iim.csic.es; 4Centro i~mar & CeBiB, Universidad de Los Lagos, Casilla 557, Puerto Montt 5480000, Chile; patricio.diaz@ulagos.cl

**Keywords:** *Alexandrium minutum*, light, paralytic shellfish poisoning, harmful algal blooms, pulse amplitude modulation, encystment

## Abstract

The changes in the cell physiology (growth rate, cell size, and cell DNA content), photosynthetic efficiency, toxicity, and sexuality under variable light and nutrient (phosphates) conditions were evaluated in cultures of the dinoflagellate *Alexandrium minutum* obtained from a red tide in the Ría de Vigo (NW Spain). The cells were grown at low (40 and 150 µE m^−2^ s^−1^), moderate (400 µE m^−2^ s^−1^), and high (800 µE m^−2^ s^−1^) light intensities in a medium with phosphate (P+) and without (P−). Cultures were acclimated to the irradiance conditions for one week, and the experiment was run for ~1 month. The cell size and DNA content were monitored via flow cytometry. Two different clonal strains were employed as a monoculture (in a P− or P+ medium) or, to foster sexuality and resting cyst formation, as a mixed culture (only in a P− medium). *A. minutum* growth was favored by increasing light intensities until 400 µE m^−2^ s^−1^. The DNA content analyses indicated the accumulation of S-phase cells at the highest light intensities (400 and 800 µE m^−2^ s^−1^) and therefore the negative effects on cell cycle progression. Only when the cells were grown in a P− medium did higher light intensities trigger dose-dependent, significantly higher toxicities in all the *A. minutum* cultures. This result suggests that the toxicity level is responsive to the combined effects of (high) light and (low) P stress. The cell size was not significantly affected by the light intensity or P conditions. The optimal light intensity for resting cyst formation was 150 µE m^−2^ s^−1^, with higher irradiances reducing the total encystment yield. Encystment was not observed at the lowest light intensity tested, indicative of the key role of low-level irradiance in gamete and/or zygote formation, in contrast to the stressor effect of excessive irradiance on planozygote formation and/or encystment.

## 1. Introduction

Harmful algal blooms (HABs) exert substantial adverse effects on fisheries, aquaculture, water quality, and public health [1,2,3]. The global increase in their frequency and severity has resulted in significant socioeconomic and sanitary impacts in coastal regions [4]. The bloom-forming dinoflagellate *Alexandrium minutum* is responsible for paralytic shellfish poisoning (PSP) in Western Europe. Global warming is expected to lead to 30 more days a year in which the ambient conditions favor *Alexandrium* blooms [5]. Studies on how the projected changes in the ambient conditions will affect the growth and virulence (toxicity and recurrence) of HAB species such as *A. minutum* are therefore needed.

The dynamics of the coastal inlets comprising the Galician Rías have been well studied and are influenced by estuarine circulation and coastal upwelling. Álvarez-Salgado et al. [6] showed that the increased residence time of waters in the southern Rías Baixas occurred in parallel with a decreasing trend in the upwelling index and found a pronounced impact of anthropogenic terrestrial fertilization. These factors strongly affect the succession regimes of phytoplankton and thus the development of HABs [6,7,8]. Indeed, the occurrence of *A. minutum* and PSP events in the southern Rías Baixas (NW Spain) has increased in recent years, the product of higher temperatures, stratification of the water column, moderate positive upwelling conditions, and the riverine inputs in late spring and summer [8,9,10]. The weakening of the coastal upwelling along the northwest Iberian shelf in recent decades has been attributed to the changing climate conditions [6,7,9]. While no significant interannual trends characterized the period between 2005 and 2021, positive anomalies were recorded between 2016 and 2021 [11].

Among European Union (EU) Member States, Spain is the second-largest producer of marine aquaculture products (2.71 × 10^5^ tons per year). It is also the world’s third-largest mussel producer and the most important mussel-producing area in the EU [12,13]. In Galicia, many jobs are directly linked to the shellfish industry (producers, depurators, processors, mussel raft (bateas) builders) and are therefore affected by the blooms of toxin-producing dinoflagellates. A PSP event in Galicia caused by *A. minutum* (1 × 10^7^ cell L^−1^) was first documented in May 1984 [14]. In subsequent years, moderate concentrations (1–5 × 10^4^ cell L^−1^) of *A. minutum* were detected in the Ría de Vigo. Following the implementation of a monitoring program by INTECMAR (Instituto Tecnolóxico para o Control do Medio Mariño de Galicia) in the 1990s, populations of *A. minutum* and/or its cysts were determined to be permanent members of the phytoplankton community of Ría de Vigo, especially in the southern Bay of Baiona (Figure 1) [14]. Within the Ría de Vigo itself, dinoflagellate cyst assemblages are dominated by other genera, and *Alexandrium* is rare or even absent [15].

One of the climate variables affected by global warming is solar irradiance, the intensity and duration of which strongly impact the growth rates of phytoplankton and the build-up of their populations. In general, the growth rate of a phototrophic dinoflagellate increases with increasing photon flux density until saturation is reached [15], with, larger cells usually associated with lower photoinhibition under excessive light, and smaller cells with higher photosynthetic rates under limiting light [16,17]. Several studies of *Alexandrium* have reported a species-specific effect of irradiance on growth. For example, the growth rate (μ) of *A. catenella* was shown to level off at 90 µmol photons m^−2^ s^−1^ [18] whereas that of *A. tamarense* increased with increasing irradiance from a blue LED, but was inhibited by fluorescent light with an irradiance <30 µmol m^−2^ s^−1^ and yellow and red LED light at irradiances <70 µmol photons m^−2^ s^−1^ [19]. *A. fundyense* reached the maximum growth at an irradiance of >120 µmol photons m^−2^ s^−1^ [20], while the highest cell concentrations of *A. pseudogonyaulax* occurred at 200 μmol photons·m^−2^·s^−1^ and the lowest at 50 μmol photons·m^−2^·s^−1^ [21]. The growth rate of a Malaysian strain of *A. minutum* increased with increasing irradiance (10, 40, 60, 100 μmol photons·m^−2^·s^−1^), with no significant difference within the range of 60–100 μmol photons·m^−2^·s^−1^ [22]. In a Chinese strain of *A. minutum* under continuous light exposure (L/D 24:0), there was no significant change across a range of 80–370 µmol photons m^−2^ s^−1^ [17]. However, for nearly all *Alexandrium* species, little is known about the effects of light on their toxin levels or sexuality rates, both of which are crucial to explaining the severity and recurrence of toxic blooms.

*A. minutum* undergoes both sexual and asexual modes of reproduction, accompanied by changes in the DNA content (haploid–diploid and haploid–haploid, respectively) [23,24]. While growth is mainly the product of mitotic division, recent studies have shown that sexual reproduction, which may include the production of large numbers of resting cysts, is also an important contributor [25]. The ability to shift between these two modes of reproduction is critical to the survival of *A. minutum*, as it enables its adaptation to fluctuations in the environment, with rapid mitotic proliferation under optimal conditions and long-term persistence in the form of (diploid) resting cysts under adverse conditions [26,27,28]. Indeed, resting cysts play major roles in bloom recurrence and geographical spread; e.g., see the reviews by [26,29,30].

Environmental factors are key regulators of the growth of *A. minutum*. DNA replication, and thus the cell cycle, follows a circadian rhythm, with the S phase occurring solely during the light period [31]. However, little is known about the effect of light intensity on the sexuality of *A. minutum*, although short photoperiods were shown to be associated with higher encystment rates, e.g., [32]. The availability of phosphate (P) is also important for dinoflagellate growth [33,34], with P deficiency related to increased sexual induction (gamete fusion) and higher meiotic rates in some species [35], including *A. minutum* [25]. Toxin production is also affected by the P levels [36], evidenced by the increases in toxin production in several species, such as *A. tamarense*, *A. minutum*, *A. catenella*, and *Karenia brevis*, under P limitation [37,38,39,40,41].

The main objective of this work was to compare the physiological response of *A. minutum* in terms of growth, toxin production, and sexuality under four different light intensities and two different P conditions. Both clonal (mainly mitotic) and mitotic growth in combination with sexuality and resting cyst production were tested. Our results provide novel insights into how light intensity and P availability, both directly affected by the changing climate conditions, affect the growth, virulence, and recurrence (via sexual resting cyst storage) of *A. minutum*. This information will contribute to the improved parameterizations in the models aimed at predicting HABs.

## 2. Materials and Methods

### 2.1. Stock Culture Conditions

The two strains of *A. minutum* (VGO 1424 and VGO 1439) used in our study were obtained from the CCVIEO Culture Collection of the Oceanographic Centre of Vigo (IEO-CSIC: https://vgohab.com/en/coleccion-de-cultivos/). These non-axenic strains were isolated from a red tide in the Ría de Vigo (northwest Spain, Figure 1) in the summer of 2018 and subsequently maintained at 19 °C in L1 medium [42] without silicates (L1_-Si_). The photoperiod was set at 12:12 (light/dark), with an irradiance during the light period of 90 µmol photons·m^−2^·s^−1^. The medium was prepared using filtered (0.22 µm pore size) and autoclaved Atlantic seawater, and the salinity was adjusted to 32 via the addition of sterile distilled water (pH 8.00 ± 0.02). Culture stocks were maintained in 2 L Erlenmeyer flasks.

### 2.2. Light Experimental Design

The experimental treatments included four irradiance levels (40, 150, 400, and 800 µE m^−2^ s^−1^); the salinity (32), temperature (19 °C), and photoperiod (12:12) were held constant (Figure 2). The cultures (400 mL of 6 × 10^3^ cells mL^−1^ in 1 L Erlenmeyer flasks) were acclimated to the different irradiance levels for one week and then divided into three replicates, each consisting of 400 mL of a 1 × 10^3^ cells mL^−1^ culture in a 1 L flask. A consistent temperature (accuracy 0.1 °C) across the different irradiances was maintained by placing the flasks in an incubator chamber (EQUITEC EGCHS 755/3) with a sunlight spectrum to provide irradiation and a forced air ventilation system to ensure a uniform horizontal distribution of the air flow.

The experimental setup lasted ~4 weeks and included assays in which the two strains were grown (i) in separate monospecific cultures, either in a replete medium (Clonal) or under a P deficiency (Clonal_-P_) and (ii) as a mixed culture under a phosphate deficiency (Cross_-P_). The P-free medium was employed to induce sexuality.

### 2.3. Growth Rates and Cyst Production

Every 2–3 days, one sample per every replicate of each irradiance was collected and fixed with acidic Lugol’s solution (0.5%) to determine the cell density. The cell suspension (minimum of 400 cells per sample) was placed in a 1 mL Sedgewick Rafter counting chamber (Pyser Optics, Kents, UK) and counted at a 100× magnification using a Zeiss inverted microscope. In the Cross_-P_ treatment, the induction of sexuality resulted in cyst formation. A sterile cell scraper dragged along the bottom and sides of the flasks was used to remove the cysts for counting as described above. At least 400 cells and all cysts were counted in every sample.

Growth rates (μ, day^−1^) were estimated following Guillard [38]: μ = (lnN_2_ lnN_1_)/(t_2_−t_1_), where N_1_ and N_2_ are cell densities (cell mL^−1^) at time t_1_ and t_2_ (days). Here, t_1_ is the day when the initial phase growth phase began, and t_2_ is the end of the exponential growth phase.

The irradiance was provided using light-emitting diodes (LEDs), due to their economic cost, energy efficiency, and durability. Cool-white light and four irradiances (40, 150, 400, and 800 µE m^−2^ s^−1^) were used in the experiments.

### 2.4. Photosynthesis Efficiency

The maximum light utilization efficiency of photosystem II (*Fv*/*Fm*) was measured in replicates of each irradiance one week after the start of the experiments using a pulse amplitude modulated fluorometer (Phyto-PAM) controlled by the WinControl-3 Windows software v2.08 (Walz, Effeltrich, Germany) with 2 mL sample volumes.

### 2.5. Toxin Analysis

One sample per every replicate of each irradiance was collected every 4–5 days to determine its toxin content. For Clonal and Clonal_-P_ samples, the volumes ranged from 20 to 40 mL, and for Cross_-P_ samples, 120–164 mL. All the samples were filtered through glass-fiber filters (GF/C 25 mm diameter; Whatman). Each filter was placed in an Eppendorf tube to which 750 μL of 0.05 M acetic acid was added. The tubes were sonicated for 1 min at 50 W and then kept at –20 °C until the analysis. The toxins were extracted by centrifuging the thawed samples at 17.978 g at 10 °C for 10 min using a Sigma 3-16KL (Sigma Laborzentrifugen GmbH, Osterode am Harz, Germany) Sartorius centrifuge. The supernatants were collected in clean Eppendorf tubes, and the extraction was repeated. The two supernatants were combined (resulting in a final volume of 1500 μL) and filtered through 0.22 μm PTFE syringe filters before they were analyzed via high-performance liquid chromatography.

Paralytic shellfish toxins (PSTs) were determined using liquid chromatography post-column oxidation followed by LC-PCOX-FLD, as reported by Rourke et al. [43], with the modifications implemented by Salgado et al. [44]. Gonyautoxins (GTXs) were chromatographically separated using a Waters XBridge^®^ Shield RP column (4.6 × 150 mm, 3.5 µm) and an injection volume of 20 µL. An isocratic mixture of 95% solvent A and 5% solvent B was run at a flow rate of 1 mL min^−1^ for 20 min. Certified reference PST standards were obtained from the CIFGA Laboratory (Spain). The LOD (s/n = 3) and LOQ (s/n = 10) for GTX4 were 0.152 and 0.496 µg mL^−1^, respectively. The sample toxin content was expressed as pg GTX4 cell^−1^.

### 2.6. Sexuality and DNA Content Analyses

Every 4–5 days, 25 to 30 mL samples were collected from each replicate of each irradiance between 10:00 and 11:00 h (UTC+1) and fixed with 20% formaldehyde (final concentration of 1%) for 24–48 h. The samples were then centrifuged at 5000× *g* for 5 min at 10 °C. For chlorophyll extraction, the pellets were transferred to an Eppendorf tube, resuspended in 1.5 mL of cold methanol, and incubated for at least 12 h at 4 °C. The samples were then centrifuged as above, and the pellets were washed in PBS (pH 7, Sigma-Aldrich, St. Louis, MO, USA), homogenized via vortexing, centrifuged as above, resuspended in 300 μL of propidium iodide (at 60 μg mL⁻^1^) and 30 μL of RNase A (at 100 mg mL⁻^1^) each, and incubated for at least 2 h at room temperature. The samples were then analyzed on a sorting flow cytometer (model SH800Z; Sony Biotechnology Inc., Tokyo, Japan) equipped with a laser emitting at 488 nm. The cells were acquired using a 100 μm microfluidic sorting chip and run at a low speed. The data were acquired in linear and log modes until at least 10,000 events had been recorded. The data analysis was conducted using the Flowing software 2.5.1. (Perttu Terho, University of Turku and Åbo Akademi University, Finland). The working template generated for sexuality analyses included three populations: C, 2C, and 4C, representing the amount of DNA present in the haploid chromosome set, the diploid chromosome set, and following the duplication of 2C in the S phase, respectively. The analysis included n ≥ 3000 cells, with an exception of n = 730 and a maximum of 55,200 cells. The working template generated for the cell size estimations included one population of *A. minutum* and was carried out using the forward scatter (FSC) parameter to obtain a relative measure that was proportional to the diameter of the cell.

### 2.7. Statistical Analysis

The statistical analyses included non-parametric Mann–Whitney U and Kruskal–Wallis tests as well as Dunn’s non-parametric all-pairs comparison post hoc test. Differences were considered statistically significant when *p* < 0.05, with a significance level of α = 0.05.

A generalized linear model (GLM) with a logit-link function for negative binomial distribution [45] was implemented using the “MASS” library [46] to identify the influence of different environmental factors on the cyst production. The light intensity (discrete variables) and number of days of the experiment (continuous variable) were tested as fixed effects.

Interaction terms were not used because they can obscure the effects of individual predictor variables [47]. The significance of each explanatory variable was determined using a x^2^ test [48]. The assumptions in the linear models and overdispersion were assessed following the methodology implemented in the R “DHARMa” library [49].

The analyses were performed using the R 4.2.1 statistical and programming software [50] and the above-mentioned packages, available through the CRAN repository (www.r-project.org/, accessed on 10 March 2024).

## 3. Results

### 3.1. Growth Rates and Cell Size

The cultures growing clonally under different P levels (Clonal and Clonal_-P_) differed significantly in their growth under the different levels of irradiance.

The different irradiance levels had no significant effects on the growth of Cross_-P_ (Figure 3). By contrast, for Clonal and Clonal_-P_, significant differences in growth occurred in response to the different irradiance levels (*p* = 0.015–4.90 × 10^−8^) (Figure 3A–D). Specifically, at the irradiances of 40 and 150 µE m^−2^ s^−1^, the growth of Clonal_-P_ was significantly higher while the Clonal cultures grew best at the irradiances of 400 and 800 µE m^−2^ s^−1^. In fact, while at 800 µE m^−2^ s^−1^, the Clonal cultures reached their highest cell densities (60,000 cells mL^−1^) from day 9 onwards, the average densities of the Clonal_-p_ cultures at the same irradiance never reached 10,000 cells mL^−1^ during the 26 days of the experiment (Table 1).

The differences (*p* = 3.03 × 10^−5^–4.83 × 10^−13^) between Clonal_-P_ and Cross_-P_ at all the irradiances were also significant (Figure 3C–E). Thus, higher cell densities were reached by Cross_-P_ at 40 and 150 µE m^−2^ s^−1^ and by Clonal_-P_ at 400 and 800 µE m^−2^ s^−1^. Under an irradiance of 40 µE m^−2^ s^−1^, Clonal_-P_ did not exhibit measurable growth, as the cell numbers remained at ~1000 cells mL^−1^ throughout the experiment, while Cross_-P_ grew until day 16, reaching ~4500 cells mL^−1^ (Table 1).

Given the very low growth at 40 and 150 µE m^−2^ s^−1^ (<5500 cells mL^−1^), the analyses of the cell size were limited to the cultures under the irradiances of 400 and 800 µE m^−2^ s^−1^. The different irradiances did not result in significant differences in the mean cellular FSC, either within the same treatment group or between groups (Figure 4).

### 3.2. Photosynthetic Efficiency

*Fv*/*Fm* was used to estimate the photoadaptation or photoinhibition and to determine the changes in the functional state resulting from the physiological changes in the cells. Significant differences in *Fv*/*Fm* were observed between the cultures grown in the presence and absence of P (*p* = 0.001). For the cultures grown in a P-containing medium, the *Fv*/*Fm* values were lower at the lowest irradiance (40 m^−2^ s^−1^ µE), whereas for the cultures grown in the absence of P, the *Fv*/*Fm* values were lower at the highest irradiance (800 µE m^−2^ s^−1^) (Table 2).

The differences in *Fv*/*Fm* according to the light intensity were highly significant (*p* = 0.0001) for the Clonal and Clonal_-P_ groups. For the Clonal cultures, the *Fv*/*Fm* values were the highest at 400 µE m^−2^ s^−1^ followed by 150 and 800 µE m^−2^ s^−1^. For the Clonal_-P_ cultures, the *Fv*/*Fm* values were the highest at 40 and 150 µE m^−2^ s^−1^ followed by 400 µE m^−2^ s^−1^ (Table 2).

None of the differences in the *Fv*/*Fm* values of the cultures grown in the absence of P were significant (Table 3); for Clonal_-P_ and Cross_-P_, the highest values were measured at the irradiances of 40 and 150 µE m^−2^ s^−1^.

### 3.3. Toxin Production

A previous analysis of the toxins produced by the strains VGO 1424, VGO 1439, and their cross focused on GTX4. Ben-Gigirey et al. [51] found that VGO 1424 and VGO 1439 mostly produced GTX4, with trace amounts of GTX3 for VGO 1424 and GTX2 + GTX3 for VGO 1439. In the present study, because GTX2 and GTX3 concentrations were below the LOQ, only the production of GTX4 was evaluated. The highest concentrations were produced by the two cultures grown in a P-free medium (Figure 5). The differences compared with the Clonal culture were significant (*p* < 0.001).

The light intensity had a significant effect on the GTX4 production (*p* < 0.001), with significant differences found for the Clonal_-P_ and Cross_-P_ cultures. For the former, the GTX4 production was significantly higher at 400 and 800 µE m^−2^ s^−1^ (the highest GTX4 production was at 800 µE m^−2^ s^−1^). For the Cross_-P_ cultures, in the comparisons of 40 vs. 800 µE and 150 vs. 800 µE m^−2^ s^−1^, the GTX4 production was consistently the highest at 800 µE m^−2^ s^−1^ (Figure 5).

The temporal evolution of the GTX4 production was also monitored (Figure 6). The toxin production by the Clonal strains at the two highest irradiances (400 and 800 µE m^−2^ s^−1^) declined significantly over time. For the Clonal_-P_ and Cross_-P_ cultures, both of which produced significantly higher levels of toxin than the Clonal culture, the toxin production increased until the midpoint of the experiment and then decreased.

### 3.4. Resting Cyst Production and DNA Content

The effects of the different treatments on the cell cycle were explored via cytometry. The percentages of 2C and 4C cells were sensitive to the irradiance level, not to the culture medium. For the Clonal and Clonal_-P_ cultures, the highest percentage of 2C cells occurred at 150 µE m^−2^ s^−1^ and the lowest percentage occurred at 40 µE m^−2^ s^−1^ (Table 4). For the Clonal cultures, the highest percentage of 4C cells occurred at 800 µE m^−2^ s^−1^ and the lowest percentage occurred at 40 and 150 µE m^−2^ s^−1^. For the Clonal_-P_ cultures, the highest percentage of 4C cells was detected at 400 µE m^−2^ s^−1^ and the lowest percentage was detected at 150 µE m^−2^ s^−1^ (Table 4).

The percentage of 4C cells in the Clonal_-P_ and Cross_-P_ cultures differed significantly (*p* = 0.003), with a higher percentage occurring in the latter. Resting cysts were found only in the Cross_-P_ cultures (max 198 cysts mL^−1^), not in the Clonal and Clonal_-P_ cultures (Table 4).

The GLM analysis showed that the light intensity and the number of days of cultivation contributed significantly (*p* < 0.05) to the variability of the *A. minutum* cyst production, based on a total explained deviance for the model of 62.3% (Table 5, Figure 7). The resting cyst production reached a maximum at an irradiance of 150 µE m^−2^ s^−1^ and on day 17 of the experiment (Figure 7B). It was significantly lower at higher light intensities (Figure 7A) and absent at the lowest light intensity studied (40 µE m^−2^ s^−1^).

An examination of the temporal changes in the proportions of 2C and 4C cells showed significant oscillations in the former in the Clonal and Cross_-P_ cultures (*p* = 0.03 and 0.02, respectively), whereas the proportion was stable in the Clonal_-P_ cultures (Figure 8 and Figure 9). By contrast, the proportion of 4C (putative meiotic) cells did not differ significantly between the three groups, although it was slightly higher in the Cross_-P_ culture. The histogram clearly changed over time, including wider C peaks at the higher light intensities, evidencing a larger proportion of cells in the S phase, mainly in the Clonal_-P_ and Cross_-P_ cultures (arrows in Figure 8 and Figure 9).

## 4. Discussion

Understanding the influence of human-induced climate warming on the geographical expansion and occurrence of HABs first requires studies of how the key cell functions linked with bloom formation, severity, and recurrence (i.e., cell growth, toxicity, and sexuality through resting cyst formation, respectively) are modulated by changes in the environmental factors. The focus of our study was on the effects of two such factors, light intensity and P availability. Their impacts on the dinoflagellate toxicity, growth, and sexual resting cyst formation as well as the potential implications for HABs are discussed below.

### 4.1. Optimal Light Levels for A. minutum Growth

The growth of *A. minutum* was favored by increasing irradiances until 400 µE m^−2^ s^−1^. At the highest irradiance tested (800 µE m^−2^ s^−1^), the growth rate and maximum density were the same as at 400 µE m^−2^ s^−1^ for the Clonal culture (Figure 3) and slightly lower for the Clonal_-P_ and Cross_-P_ cultures. For the latter groups, the inhibition of growth at 800 µE m^−2^ s^−1^ may have been related to the combined effect of light and P stresses.

The lowest tested irradiance mimicked the low light exposure of dinoflagellate populations inhabiting the pycnocline [52], whereas the highest tested irradiance was roughly the same as the photon flux registered at the sea surface during a sunny spring or cloudy summer day in the Galician Rías. Light intensities of up to 740 µmol m^−2^ s^−1^ have been recorded at the sea surface [53] and according to our work could inhibit dinoflagellate growth, including the bloom formation by *A. minutum*. Periods of high irradiances accompanied by stratification could result in the exposure of cells near the sea surface to light levels that exceed their photosynthetic capacity, thus inducing light stress [54,55]. This was suggested by a previous study in which high irradiances were shown to cause species-specific stress effects, with consequences for bloom formation [56]. For example, while light had no effect on the cell volume of *A. fundyense* or *A. minutum*, larger cell volumes have been reported for light-exposed dinoflagellates such as *Heterocapsa* [56].

### 4.2. Toxicity Increases with Higher Light Levels and P Deficiency

Both a role for P in the regulation of toxin metabolism and the stimulation of PSP synthesis via P stress in *Alexandrium* species, including *A. minutum*, has been observed [57,58,59,60]. Our results confirm those observations and additionally show that high light levels further contribute to the increment in cell toxicity that occurs under P deprivation. Specifically, for *A. minutum* growing under P stress, the toxicity levels were up to five times higher than the levels measured in cells growing in a P-containing medium, depending on the light intensity. The toxicity decreased over time regardless of the medium and light conditions, probably due to nutrient exhaustion. Extrapolating these results to the field, the changes in precipitation and thermohaline stratification expected under a warming climate, when accompanied by larger P losses and higher irradiance conditions, will promote *A. minutum* toxicity and growth.

### 4.3. Resting Cyst Formation Is Light-Dependent but Decreases at High Irradiances

Phosphate is the “ultimate” limiting nutrient as it has no biogenic source. A recent study suggested that P limits phytoplankton growth in the oceans more frequently than previously considered [61]. Nutritional stress has been shown to induce sexuality in dinoflagellate cultures [62], while a laboratory study showed that a P-free medium induces sexuality in *A. minutum*, by triggering cyst formation [63]. Although *A. minutum* is heterothallic, gametes and zygotes can also be produced in clonal cultures [25]. This implies that sexuality affects the growth rates of both clonal and crossed strains. In our study, the importance of sexuality was demonstrated by the stronger negative effect on growth exerted on the Cross_-P_ cultures at the highest light intensity (Figure 3E). In those cultures, sexuality was, as expected, higher than in either the Clonal or Clonal_-P_ cultures. Moreover, only in the Cross_-P_ cultures were resting cysts observed. The flow cytometry provided a snapshot of the cell cycle profile at the sampling time (10 am), when the synchronized cells had already divided and the S phase (between C and 2C peaks) was starting [31]. Therefore, the 2C peaks corresponded to zygotes (from self- or out-mating), to unsynchronized mitotic cells, or to stressed mitotic cells unable to divide. The increase in 2C cells with increasing light intensity, as observed in the histogram, was strongest at the highest irradiances (400 and 800 µmol m^−2^ s^−1^) and during P deficiency, i.e., the conditions in which more cells in the S phase were also observed although the growth rates were not higher. This histogram pattern could represent the effects of photoinhibition, as S-phase cells are more sensitive to particular types of stress. For example, high levels of turbulence provoke the accumulation of *A. minutum* cells with S-2C DNA contents, i.e., cells that are unable to divide until favorable conditions are restored [64].

Our study also showed that light affects the resting cyst formation. The resting cyst formation was highest at 150 µE m^−2^ s^−1^ but low at higher light intensities (Figure 7) and was not observed at the lowest light intensity tested. Two consecutive steps are needed for resting cyst formation that could be affected by light: (i) zygote formation and (ii) zygote encystment. While, to our knowledge, no studies so far have analyzed how light affects zygote formation and/or encystment, shorter photoperiods are known to induce a higher encystment in some dinoflagellates [32,65]. That a minimum light threshold is necessary for zygote formation whereas high light levels limit planozygote encystment therefore seems plausible. Nonetheless, resting cysts were formed at all irradiances except the lowest one tested. Since even low levels of resting cyst formation may suffice to ensure bloom recurrence and species expansion [66], whether the negative effect of light on encystment is a relevant limiting factor for bloom development is unclear. Another open question is whether a high light intensity affects encystment or planozygote formation. The answer is crucial to determining the consequences for *A. minutum* blooms. Other stressors, such as parasites, do not affect zygote formation but instead promote zygote division (i.e., meiosis) over encystment [67], thus accelerating sexual recombination since *A. minutum* resting cysts have a mandatory dormancy period of more than 1 month [63] before germination and meiosis can occur. Therefore, a lower rate of encystment is not always the consequence of a lower rate of sexuality and indeed, could imply the opposite.

## 5. Conclusions

The current climate trends are expected to disrupt both the human activities and ecosystems in coastal areas. In the Southern Rías Baixas, PSP outbreaks, shellfish harvesting bans due to *A. minutum* outbreaks, and extensive red tide events have become increasingly common. The ability to predict and mitigate the dinoflagellate blooms requires an understanding of the environmental and biological factors behind these events. Our study showed that *A. minutum* thrives under the high levels of irradiance that seasonally occur in the Rías Baixas while P deficiency limits growth but induces encystment. Moreover, the interaction of high irradiances and low P levels results in large increases in the GTX4 content of *A. minutum* proliferations. Resting cyst formation, by contrast, was found to be negatively affected by light levels above a relatively low threshold. The irradiance-dependent accumulation of resting cysts after *A. minutum* proliferations could help to explain the recurrence of *A. minutum* blooms in the Rías Baixas.

The increasing irradiance and decreasing P levels resulting from the current climate trends favor the increases in the density and toxicity of *A. minutum* blooms. However, higher light levels could also reduce population encystment and thereby limit the capacity for bloom recurrence. Nonetheless, at higher irradiances, resting cysts were still produced, such that whether the reduction in encystment was strong enough to prevent bloom recurrence remains unclear. Identifying the light-sensitive steps of the dinoflagellate sexual cycle is essential to understanding how the environmental and physiological factors interact to influence the development of HABs. The ability of high irradiances and P limitation to increase the bloom severity suggests that climate change will increase the frequency and virulence of dinoflagellate blooms.

## Figures and Tables

**Figure 1 microorganisms-12-01480-f001:**
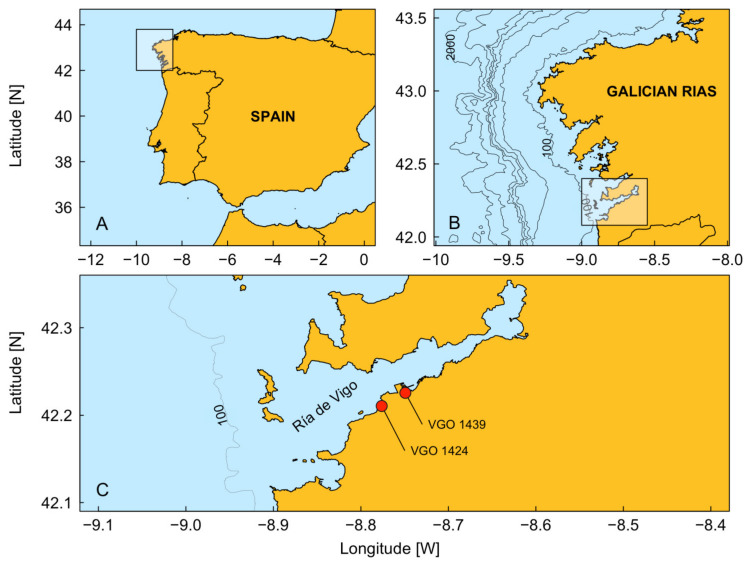
Map of the study area, showing (**A**) the Iberian Peninsula, (**B**) the Galician Rías in NW Iberia (rectangle indicates Ría de Vigo), and (**C**) the location of the two sampling stations (red circles) where *A. minutum* strains (VGO 1424 and VGO 1439) were isolated.

**Figure 2 microorganisms-12-01480-f002:**
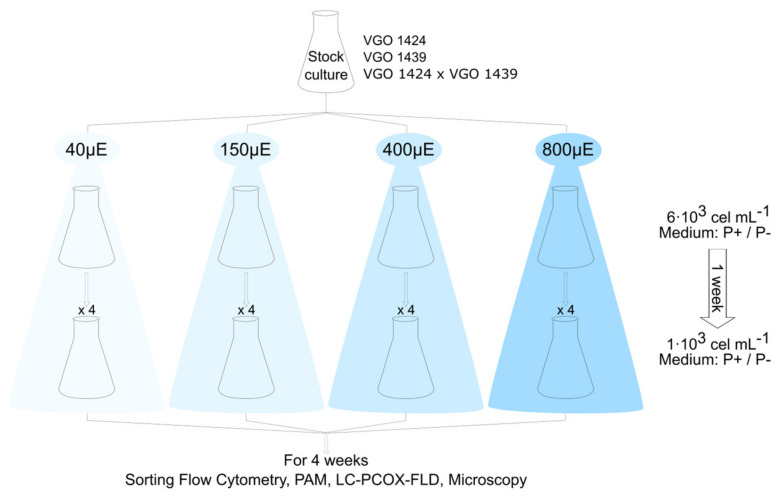
Experimental design scheme. The cultures consisted of strains VGO 1424, VGO 1439, and a mixture of both (VGO 1424 × VGO1439). Light irradiance was set at 40, 150, 400, and 800 µE m^−2^ s^−1^. The cultures were grown in medium L1_-Si_ (P+) or L1_-Si_ without phosphate (P−). The cell concentration at the start of the 1-week acclimation was 6 × 10^3^ cells mL^−1^. The starting cell concentration thereafter was 1 × 10^3^ cells mL^−1^ (4 replicates). During a ~4-week period, the cultures were analyzed via sorting flow cytometry, Pulse Amplitude Modulation (PAM), liquid chromatography post-column oxidation followed by florescence detection (LC-PCOX-FLD) and microscopy.

**Figure 3 microorganisms-12-01480-f003:**
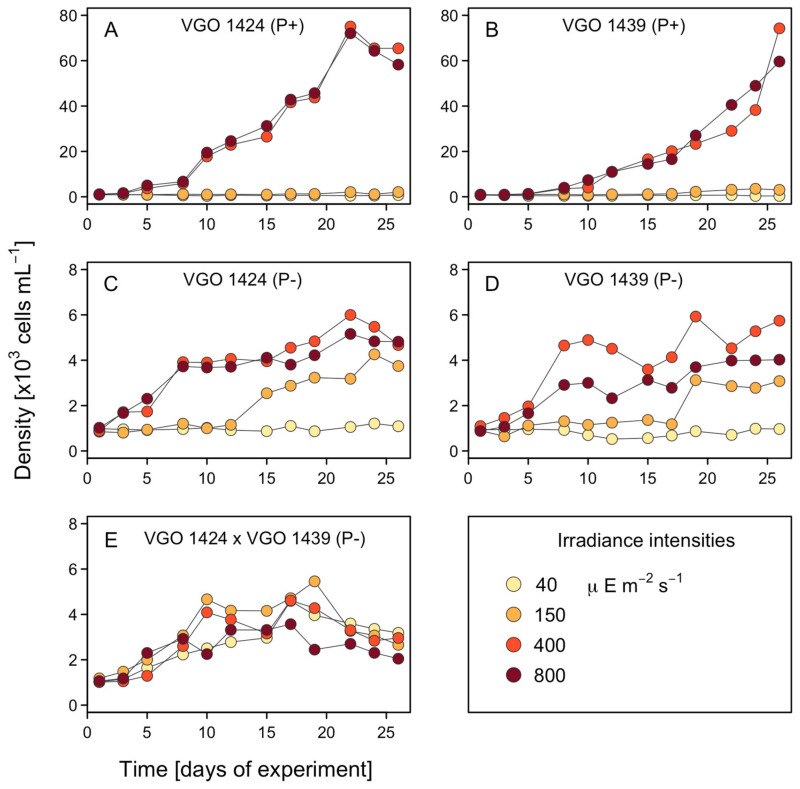
Growth rates of *A. minutum* (cell densities: ×10^3^ cells mL^−1^) during the 26-day experiment under four irradiance intensities. (**A**,**C**) VGO 1424; (**B**,**D**) VGO 1439; (**E**) VGO 1424 × VGO 1439 cross. Medium L1_-Si_ with phosphate (P+) or L1_-Si_ without phosphate (P−).

**Figure 4 microorganisms-12-01480-f004:**
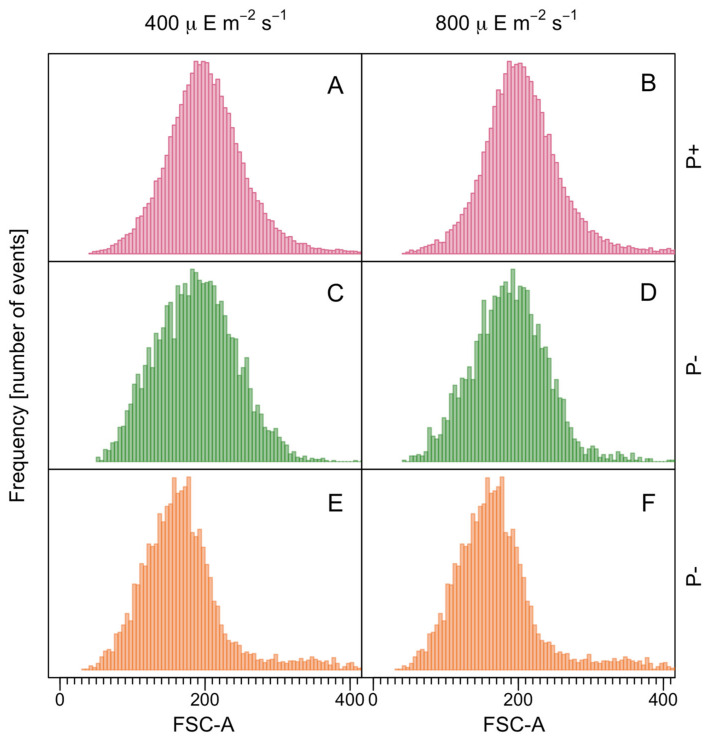
Flow cytometry histogram of forward scatter (FSC-A) in a comparison of cell size. (**A**,**B**) Clonal (P+); (**C**,**D**) Clonal_-P_ (P−); (**E**,**F**) Cross_-P_ (P−).

**Figure 5 microorganisms-12-01480-f005:**
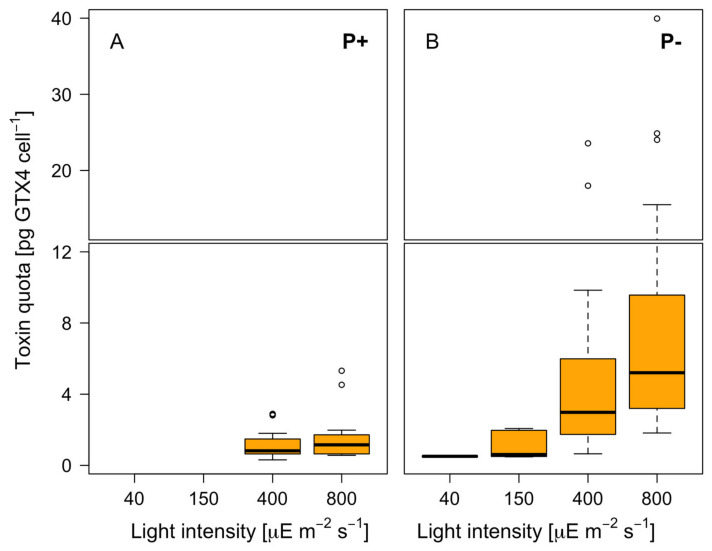
Toxin quota (pg GTX4 cell^−1^) at the tested light intensities. (**A**) Clonal (P+); (**B**) Clonal_-P_ and Cross_-P_ (P−).

**Figure 6 microorganisms-12-01480-f006:**
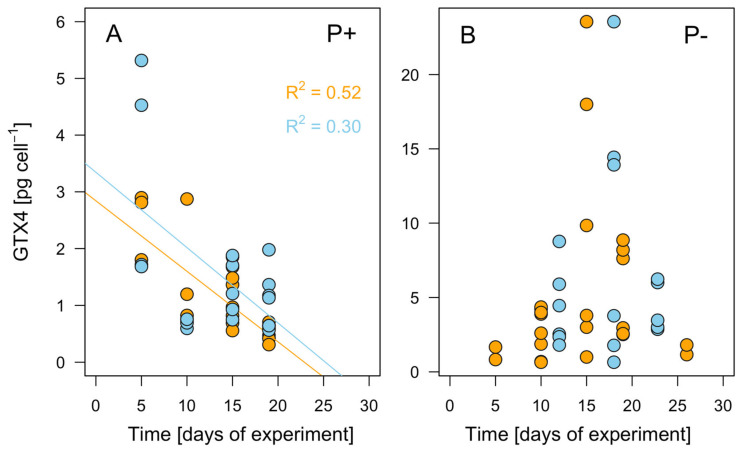
Toxin rates (pg GTX4 cell^−1^) over time at light intensities of 400 (blue) and 800 (orange) µE m^−2^ s^−1^. (**A**) Clonal (P+); (**B**) Clonal_-P_ and Cross_-P_ (P−).

**Figure 7 microorganisms-12-01480-f007:**
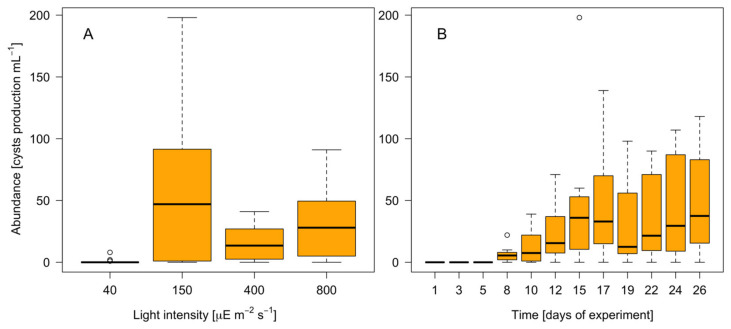
Resting cyst formation in the Cross_-P_ culture according to the tested light intensities (**A**) and experimental time (**B**).

**Figure 8 microorganisms-12-01480-f008:**
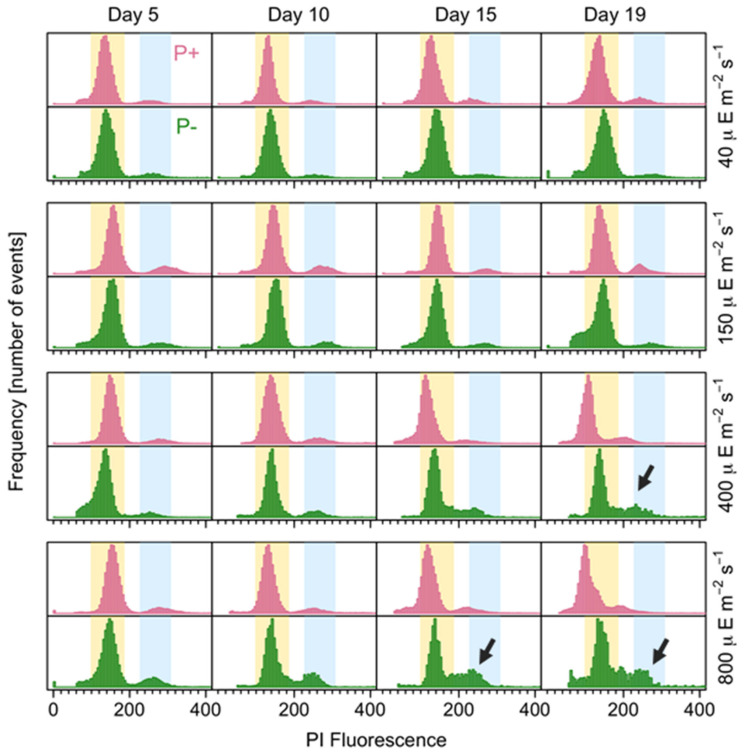
DNA fluorescence of propidium iodide-stained cells, showing the proportion of C (yellow shaded area) and 2C (blue shaded area) cells over time in strains growing clonally with (pink) and without (green) phosphate and at different irradiances. Arrows point to a notably wider C to 2C (S) phase.

**Figure 9 microorganisms-12-01480-f009:**
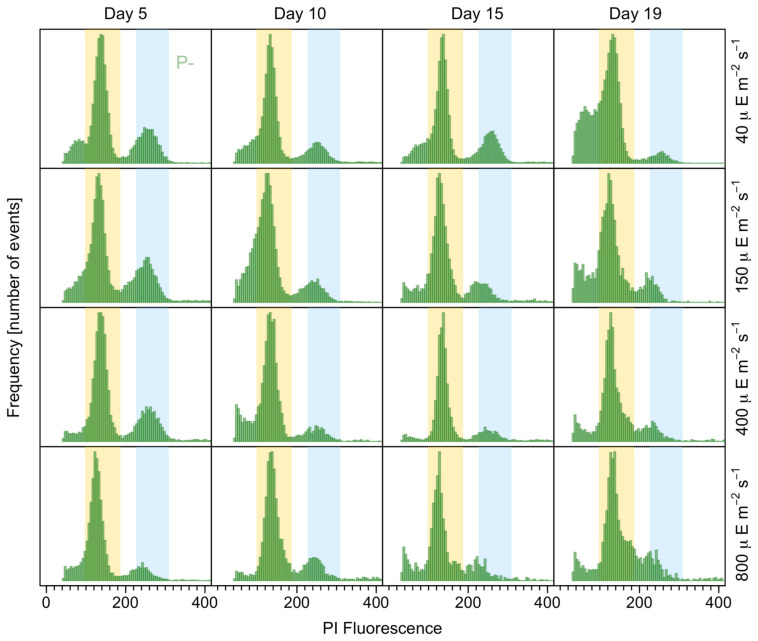
DNA fluorescence of propidium iodide-stained cells, showing the proportion of C (yellow shaded area) and 2C (blue shaded area) cells over time in the Cross_-P_ culture (sexual) at different irradiances.

**Table 1 microorganisms-12-01480-t001:** Maximum cell density (cells mL^−1^) reached at each irradiance level (40, 150, 400, and 800 µE m^−2^ s^−1^) by the Clonal (VGO 1424 and VGO 1439 in L1_-Si_ medium), Clonal_-P_ (VGO 1424 and VGO 1439 cultures in phosphate-free L1_-Si_ medium), and Cross_-P_ (VGO 1424 × VGO 1439 in phosphate-free L1_-Si_ medium) cultures at the end of the 26-day experiment.

Light Intensity (µE m^−2^ s^−1^)	Clonal	Clonal_-P_	Cross_-P_
40	830	1095	4595
150	2580	3521	5458
400	69,856	3580	4613
800	58,950	4572	3570

**Table 2 microorganisms-12-01480-t002:** *Fv*/*Fm* values of the Clonal, Clonal_-P_, and Cross_-P_ cultures at each light intensity during 8 days of exposure.

Light Intensity (µE m^−2^ s^−1^)	Clonal *Fv*/*Fm*	Clonal_-P_ *Fv*/*Fm*	Cross_-P_ *Fv*/*Fm*
40	0.298	0.705	0.722
150	0.473	0.723	0.583
400	0.592	0.591	0.564
800	0.427	0.353	0.302

**Table 3 microorganisms-12-01480-t003:** *p*-values (the bold values denote the statistical significance at the *p* < 0.05 level) of the *Fv*/*Fm* values at each light intensity determined for the Clonal vs. Clonal_-P_ and Clonal_-P_ vs. Cross_-P_ cultures.

Light Intensity	Clonal vs. Clonal_-P_	Clonal_-P_ vs. Cross_-P_
40	**0.005**	0.402
150	**0.005**	0.065
400	0.818	0.429
800	0.261	0.643

**Table 4 microorganisms-12-01480-t004:** Maximum percentages of 2C and 4C cells and maximum cyst production by strains VGO 1424, VGO 1439, and VGO 1424 × VGO 1439 when cultured in a phosphate-free L1_-Si_ medium (P−). The standard deviations (SD) of the maximum percentages of cells with 2C and 4C DNA contents and maximum cyst production are reported.

	% 2C	SD (% 2C)	% 4C	SD (% 4C)	Cysts (Cyst mL^−1^)	SD Cyst (Cyst mL^−1^)
VGO 1424 (P−)	7.96	1.66	3.04	0.72	0	0
VGO 1439 (P−)	9.63	3.19	1.79	0.52	0	0
Cross (P−)	13.95	3.99	4.49	1.24	198	39.73

**Table 5 microorganisms-12-01480-t005:** Statistical significance (*x*^2^ test [48]) of the contribution of each variable to the variability of *A. minutum* cyst abundance, determined using a generalized linear model with a logit-link function for the negative binomial distribution. Predictive variables were light intensity and days of the experiment. Significant effects (*p* < 0.05) are indicated with an asterisk.

Predictive Variable	Degrees of Freedom (Df)	Deviance	Residual Df	Residual Deviance	*p* (>*X*^2^)
Null			95	274.23	
Light intensity	3	110.706	92	163.53	<0.0001 *
Days of the experiment	1	61.682	91	101.84	<0.0001 *

## Data Availability

The original contributions presented in the study are included in the article, further inquiries can be directed to the corresponding authors.

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
