# Peer review of "Climate Change Stressors, Phosphate Limitation, and High Irradiation Interact to Increase Alexandrium minutum Toxicity and Modulate Encystment Rates"

_microorganisms, 2024, doi:10.3390/microorganisms12071480_

Round 1

Reviewer 1 Report

Comments and Suggestions for Authors

please check the attachment.

Comments on the Quality of English Language

Minor spell check and language revision are required.

Author Response

Replies to specific comments:

  1. Define abbreviation when scientific names are first cited (PAM, GTX4, …)

The following abbreviations have been defined when they first appeared in the text:

EU – European Union

L:D – Light:dark

LC-PCOX-FLD – Liquid Chromatography Post-Column Oxidation followed by Florescence Detection

HPLC – High Performance Liquid Chromatography

GTXs – Gonyautoxins

  1. Avoid abbreviations in keywords section

The keyword “PAM” has been changed to “Pulse Amplitude Modulation” to avoid abbreviations.

  1. Line 40: Cite the reference properly 

The reference in the text “see review by 3 and references there in” has been changed to “[3]”.

  1. For all parameters tested indicate the repetition number

Initial cultures were acclimated to 40, 150, 400 and 800 µE m–2 s–1 respectively for one week in 1L Erlenmeyer flask (one Erlenmeyer flask per irradiance containing 400 mL of 6 ´ 103 cells mL-1). Before that, every flask was divided into 3 replicates in 1L flasks with 400 mL of 1x103 cells mL-1 each.

For the parameters tested the repetition number was:

  • Growth rate and cyst production: every 2-3 days, one sample per every replicate of each irraiance was collected to determine cell density and cyst cuantification. Previous studies in the group have shown that the strains used in this study are heterothallic. Thus, only in Cross-P cysts were counted.
  • Photosynthesis efficiency: one sample per every replicate of each irraiance was collected to determine Fv/Fm once one week after beginning the experiments.
  • Toxins analysis: every 4-5 days, one sample per every replicate of each irradiance was collected to determine toxin contents
  • Sexuality and DNA content analyses: every 4-5 days, one sample per every replicate of each irradiance was collected to determine DNA content, to study sexuality and cell measurement.

For all parameters tested, it has been added in the text that a sample of X volume was collected from each of the three replicates for the analysis of each parameter every Y days.

  1. Minor spell check and language revision are required.

In response to your comment regarding the revision of the English language, the manuscript has been submitted to a professional editor to correct any spelling and grammatical errors.

  1. Extend the conclusion part to deeply state the importance of the obtained results

To comprehensively present the findings of this study, the section dedicated to final conclusions has been expanded.

Reviewer 2 Report

Comments and Suggestions for Authors

This is an interesting paper on how combined exposure to (low and high) light and (low and high) phosphorus stress affects the growth, cyst formation and toxicity of the dinoflagellate Alexandrium minutum. This protozoan species is known to be the species responsible for paralytic shellfish poisoning in Western Europe. Modern approaches to experimental planning and the cytometry method were used. The results confirm the key role of these factors and their combination in the formation of gametes and/or zygotes, which may be of interest to a wide scientific audience and further developed in new studies.

The relevance of the work is determined by the authors in close link with climate warming, since climate factors strongly affect the life characteristics of dinofllagellates, succession regimes of phytoplankton and the occurrence of harmful algal blooms.

In Introduction part it is need to add some sentences about typical dinoflagellate life cycle. For example as it was present in Díaz & Figueroa, 2023 

Line 72. One of the variables affected by global warming is solar irradiance, that influences reproductive rates and toxic virulence... of... (add in relation to what..., whose parameters....)

When describing previous research, you must use the past tense. See Line 80. Several Alexandrium studies have emphasized the effect of irradiation on growth, a species-specific response.

The methodological part in general is clear to me. The statistical analysis was chosen correctly. One question arose regarding this part. Line 166. Justify the choice of these values ​​range of 4 irradiances (40, 150, 400 and 800) Did you proceed from preliminary tests?

Lines 255-256 belong to the methodological part, perhaps it would be better to move them there.

Line 229 "significative" differences should be corrected as "significant"...

The results did not raise any doubts in me, they are described in clear language, with statistical tests of differences between options and observations... However, usually in articles it is necessary to indicate options with statistical differences in graphs/tables with an asterisk or letters (Fig 7). Although the authors provide p values ​​in the text, to be completely reliable, is it possible to add all the statistics materials to the supplementary material?

Discussion. Encystation was not observed at the lowest light intensities tested, indicating a key role in gamete and/or zygote formation, although there is a stressor effect on flatzygote formation and/or encystation under excessive light. Is the inhibition of encystment detected in bright light a specific feature of the studied species or is it a nonspecific phenomenon that is also characteristic of other phytoplankton/dinoflagellate species? The same applies to the effect of low intensity on the formation of zygotes. Is this only characteristic of A. minutum? If there is relevant literature about this, please cite it.

The bibliography contains 59 works, the review of knowledge covered is extensive. I recommend adding only the following work to this list: 
Díaz, P.A.; Figueroa, R.I. Toxic Algal Bloom Recurrence in the Era of Global Change: Lessons from the Chilean Patagonian Fjords. Microorganisms 202311, 1874. https://doi.org/10.3390/microorganisms11081874

Comments on the Quality of English Language

When describing previous research, you must use the past tense. 

The text can be edited by a native speaker or a professional editor. There are places that raise questions due to linguistic nuances.

Author Response

Replies to specific comments:

  1. In Introduction part it is need to add some sentences about typical dinoflagellate life  For example as it was present in Díaz & Figueroa, 2023 

In line with the guidance provided to enhance the introduction by incorporating details of the typical life cycle of dinoflagellates, new information has been integrated into this section.

  1. Line 72. One of the variables affected by global warming is solar irradiance, that influences reproductive rates and toxic virulence... of... (add in relation to what..., whose parameters....)

The sentence " One of the variables affected by global warming is solar irradiance, that influences reproductive rates and toxic virulence " has been rewritten and information has been added to clarify the effect that changes in solar irradiance due to global warming may have on the physiology of phytoplankton. The new sentence is “One of the variables affected by global warming is solar irradiance, whose changes (light intensity, hours of sunlight) display an enormous impact in growth rates of phytoplankton and the build-up of their populations”.

  1. When describing previous research, you must use the past tense. See Line 80. Several Alexandrium studies have emphasized the effect of irradiation on growth, a species-specific response.

We have sent the manuscript to a professional editor to correct the spelling and grammar of the whole text in order to fix the error you mention and others we have made.

  1. The methodological part in general is clear to me. The statistical analysis was chosen correctly. One question arose regarding this part. Line 166. Justify the choice of these values range of 4 irradiances (40, 150, 400 and 800) Did you proceed from preliminary tests? 

Previous experience in our group with similar laboratory settings in 
other algal groups and dinoflagellate species were valuable as an 
initial clue, but could not be extrapolated or directly applied to the 
particular case of Alexandrium minutum. Thus, as mentioned in the 
discussion, the irradiances chosen as minimum and maximum in the assay 
were representative of field conditions in our study area. The 
additional two values at 150 and 400 µmol m-2 s-1 were chosen to obtain sample 
results at intermediate irradiances within that range. No additional 
light treatments could be implemented in the same experimental setup 
due to the spatial limitations into the incubator chamber.

  1. Lines 255-256 belong to the methodological part, perhaps it would be better to move them there.

As the sentence "Approximation to cell size was performed using FSC, which is proportional to the diameter of the cell " was in the results, this information has been moved to the material and methods section. The new sentence is as follows " The working template generated for cell size estimations included one population of A. minutum and was carried out using the forward scatter (FSC) parameter to obtain a relative measure that was proportional to the diameter of the cell”.

  1. Line 229 "significative" differences should be corrected as "significant"...

The error has been corrected, now the sentence is “The different irradiance levels had no significant effects on the growth of Cross-P (Fig. 3). By contrast, for Clonal and Clonal-P significant differences in growth occurred in response to different irradiance levels (ρ = 0.015–4.90 ´ 10-08) (Fig. 3A–D). Specifically, at irradiances of 40 and 150 µE m–2 s–1, the growth of Clonal-P was significantly higher while the Clonal cultures grew best at irradiances of 400 and 800 µE m–2 s–1”.

  1. The results did not raise any doubts in me, they are described in clear language, with statistical tests of differences between options and observations.. However, usually in articles it is necessary to indicate options with statistical differences in graphs/tables with an asterisk or letters (Fig 7). Although the authors provide p values in the text, to be completely reliable, is it possible to add all the statistics materials to the supplementary material?

In order to indicate the statistical differences in figure 7, table 5 has been added. In addition, the material and methods section on statistical analyses has been expanded and completed with more information on the results section. We considered this information sufficiently relevant to be included in the original manuscript and have therefore expanded the text in these sections.

  1. Encystation was not observed at the lowest light intensities tested, indicating a key role in gamete and/or zygote formation, although there is a stressor effect on flatzygote formation and/or encystation under excessive light. Is the inhibition of encystment detected in bright light a specific feature of the studied species or is it a nonspecific phenomenon that is also characteristic of other phytoplankton/dinoflagellate species? The same applies to the effect of low intensity on the formation of zygotes. Is this only characteristic of A. minutum? If there is relevant literature about this, please cite it. 

Regarding the relationship we have found between encystment and light, the literature we have found focuses on the light conditions (light or dark) and how it affects the growth rate of swimming cells and consequently the encystment peak. For example, in the case of laboratory studies with Peridinium gatunense, cyst formation was observed both in the light and in the dark and at different temperatures. However, the abundance of cysts in the experiment was very low and they were not found in all replicates, so they could not be sure whether or not cyst formation was due to light.

Alster, A., dubinsky, Z. and Zohary, T. (2006), Encystment of Peridinium gatunense: occurrence, favourable environmental conditions and its role in the dinoflagellate life cycle in a subtropical lake. Freshwater Biology, 51: 1219-1228. https://doi.org/10.1111/j.1365-2427.2006.01543.x

To our knowledge, the effect of low light intensity on zygote formation is not widely studied. Experiments on various dinoflagellate species have examined the effects of light on growth rates, toxin production, nitrate uptake, cell yield, pigment and lipid content, photosynthetic efficiency, and even ingestion rates of mixotrophic species. However, we have not found any other literature addressing this topic specifically in Alexandrium or other dinoflagellates. For this reason, we consider the findings of this work to be relevant and intend to continue studying the effects on the entire life cycle of Alexandrium minutum. In the future, we aim to investigate whether this phenomenon is common to other dinoflagellate species.

  1. The bibliography contains 59 works, the review of knowledge covered is extensive. I recommend adding only the following work to this list: Díaz, P.A.; Figueroa, R.I. Toxic Algal Bloom Recurrence in the Era of Global Change: Lessons from the Chilean Patagonian Fjords. Microorganisms 2023, 11, 1874. https://doi.org/10.3390/microorganisms11081874 

We have included the reference you suggest. In addition, to expand the information on the life cycle, we have added some more citations, and the manuscript now has 67 citations.
